# Response of an HER2-Mutated NSCLC Patient to Trastuzumab Deruxtecan and Monitoring of Plasma ctDNA Levels by Liquid Biopsy

**Markus Falk** [1], **Eva Willing** [1], **Stefanie Schmidt** [1], **Stefanie Schatz** [1], **Marco Galster** [2], **Markus Tiemann** [1], **Joachim H. Ficker** [3] **and Wolfgang M. Brueckl** [3,*]

[1] Institute for Histopathology Hamburg, Fangdieckstraße 75A, 22547 Hamburg, Germany
[2] Department of Radiology, Paracelsus Medical University, General Hospital Nurnberg, Ernst-Nathan-Str.1, 90419 Nuremberg, Germany
[3] Department of Respiratory Medicine, Allergology and Sleep Medicine, Paracelsus Medical University, General Hospital Nurnberg, Ernst-Nathan-Str.1, 90419 Nuremberg, Germany
[*] Correspondence: wolfgang.brueckl@klinikum-nuernberg.de

**Abstract:** HER2-targeted therapy is currently the subject of several studies in lung cancer and other solid tumors using either tyrosine kinase inhibitors (TKI) or targeted-antibody–drug conjugates. We describe a 61-year-old female patient with HER2 mutated adenocarcinoma of the lungs who received chemo-immunotherapy, followed by trastuzumab deruxtecan (T-DXd) and third-line Ramucirumab/Docetaxel at disease progression. Plasma ctDNA monitoring was obtained at 12 timepoints during therapy and revealed HER2 mutation allele frequencies that corresponded to the clinical course of disease. HER2-targeted T-DXd therapy resulted in a profound clinical response and may be an option for NSCLC patients carrying an activated HER2 mutation. Longitudinal liquid biopsy quantification of the underlying driver alteration can serve as a powerful diagnostic tool to monitor course of therapy.

**Keywords:** HER2 mutation; trastuzumab deruxtecan; liquid biopsy; disease monitoring

## 1. Introduction

Human epithelial growth factor receptor 2 (HER2, also known as ERBB2) is a member of the ErBB receptor tyrosine kinase family, which have been identified as oncogenic driver genes in 2–4% of non-small-cell lung cancers (NSCLC) [1]. HER2 targeting has influenced the treatment of breast and gastric cancers. Recently, a HER2-directed antibody–drug conjugate, trastuzumab deruxtecan, has been granted by the FDA for metastatic NSCLC patients who have received prior systemic therapy. Classically, patients with HER2 mutant NSCLC have been treated with standard chemo- or immunotherapy, which are known to have limited activity in second- or third-line treatment [2,3]. HER2 is a complex therapeutic target, as point mutations, protein overexpression and gene copy number variations can occur and may respond differently to HER2 inhibition. Three-dimensional modeling has demonstrated that HER2 exon 20 insertion mutations induce a constitutively active kinase conformation and are therefore regarded as activating oncogenic mutations [4]. In breast cancer, usually HER2 protein expression and HER2 gene amplification are correlated to one another, while in lung cancer this correlation appears to be weaker [5,6]. HER2 is overexpressed in 2.4–38% of lung cancer cells. HER2 overexpression and HER point mutations are not clearly associated and although both alterations can occur mutually within the same tumor, this is only rarely the case [7]. Preclinical data have demonstrated that the HER2 receptor together with the attached HER2-antibody–drug conjugate is internalized more efficiently in cells that carry an activating HER2 point mutation. This may indicate that HER2 point mutation, rather than HER overexpression, is driving the therapeutic effect of T-DXd.

Accordingly, several strategies are currently being evaluated to target HER2, including TKI, bispecific antibodies, or antibody–drug conjugates [7]. Despite the lack of large prospective studies, it is widely accepted that the response to checkpoint inhibition in tumors with driver mutations is generally poor [8,9]. T-DXd is an antibody–drug conjugate directed against HER2 that showed superior efficacy to trastuzumab emtansine in breast cancers that had progressed on trastuzumab and taxane [10]. Very recently, excellent outcome data of the DESTINY-Breast04 study on HER2 low breast cancer patients have been published [11].

Analysis of HER2 mutations is usually performed using tissue samples, but analysis based on liquid biopsies is also feasible and plasma-derived ctDNA (circulating tumor DNA) can serve as an analyte to detect low-level molecular alterations in patients with solid tumors [12]. We used a hybrid capture (HC) next-generation sequencing (NGS)-based approach to track an HER2-activating mutation that was primarily diagnosed by a different pathology on an FFPE (formalin-fixed paraffin-embedded) tissue sample using amplicon-based NGS technology.

## 2. Case Report

In April 2020, a 61year-old female patient presented with cough and increasing dyspnea that had developed within six weeks. Because antibiotics were not effective, the general practitioner arranged a CT scan of the thoracic region. The patient is a never smoker, retired school teacher and married, her father had died of lung cancer. Apart from cataract surgery, the patient had no relevant medical history. Clinical and pathological diagnosis including PET-CT and pathological evaluation was performed, which revealed a TTF1-positive adenocarcinoma in the left upper lobe with a solitary pulmonary metastasis (cT3 cN2 M1a (pul), UICC IVA). Primary tumor biopsy (FFPE) showed no PD-L1 expression (TPS 0%), but an ERBB2/HER2 deletion–insertion variant in exon 20. The p.G776delinsVC (c.2326_2327insTTT) mutation was found with an allelic frequency of 43% as determined by semi-conductor-based NGS (Ion GeneStudio™ S5 Plus System, Thermo Fisher, Waltham, MA, USA and no further alteration in a set of 50 selected genes was detected at primary diagnosis. The p.G776delinsVC is a relatively rare variant that has been described in 0.14% of NSCLC [13]. Similar to other typical exon 20 insertion variants, the p.G776delinsVC is located within the ATP-binding region of the HER2 kinase domain.

From May to July 2020, the patient received four cycles of a palliative first-line chemo-immuno combination, including pembrolizumab/pemetrexed/carboplatin according to the approved KN 189 regimen. CT scans of the thorax and abdomen revealed a mixed response in August 2020, so two more chemo-immuno therapies were planned after consolidating radiation of the right lung tumor. While the fifth course was administered on time, the sixth cycle was postponed until October due to leucopenia (CTC grade 3). Regularly performed CT scans confirmed stable disease. In May 2021, after the tumor had progressed again, targeted therapy with T-DXd was started as second-line therapy (as off-label use as the drug is not approved for this indication by the EMA) and partial remission was observed for seven months followed by clinical progression in January (Figure 1). There were no toxicities CTC grade 3 or 4 during the time of treatment with T-DXd. In addition, there was no delay or dose reduction. In January 2022, after progressive disease had occurred with new pulmonary metastases, treatment was switched to third-line ramucirumab, a VEGFR antibody in combination with docetaxel. In April 2022, after three months of ramucirumab plus docetaxel, the patient presented with a partial remission, regression of right pulmonary metastases, absence of new metastases, and near complete remission of pleural spread. However, due to significant side effects, including mucositis, nausea, epistaxis, watery eyes and hand-foot syndrome, the patient requested a three-month treatment break and is still living without treatment (June 2022).

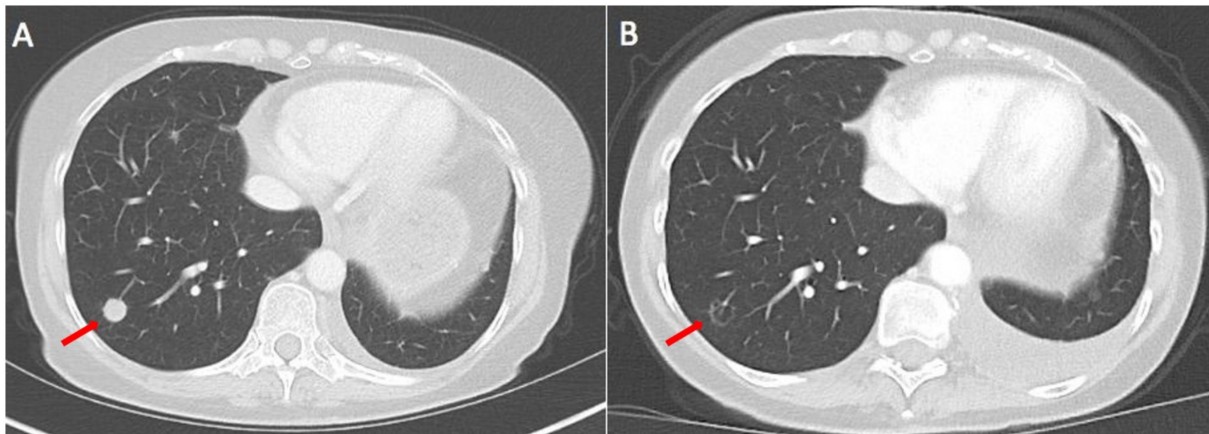

**Figure 1.** CT scans before start of T-DXd and after 5 months of treatment. One pulmonary nodule in the right lower lobe (arrow) almost diminishes. In addition, there is a small increase in pleural effusion in the left thorax (no malignant pleural effusion). (**A**) CT scan before start of T-DXd; (**B**) CT scan after 4 cycles of T-DXd; arrows indicate a lung metastasis which had diminished on Figure B.

Starting with the initiation of T-DXd in May 2021, until the end of February 2022, a total of 12 consecutive plasma samples were collected, cfDNA was extracted and analyzed via HC NGS technology using the SureSelect XT HS2 DNA Library Preparation and Target Enrichment Kit from Agilent according to the manufacturer's protocol. ctDNA was analyzed for the presence of point mutations and gene fusions in a set of 50 lung-cancer-associated genes [11]. The SureSelect XT HS2 DNA Library Preparation and Target Enrichment Kit is provided with molecular barcodes to remove false-positive events and to detect mutations with low allele frequency, necessary for liquid biopsy. For successful processing, 120 mrp (million read pairs) were required to achieve the necessary mean coverage $2500\times$ over the territory of genes of interest. ctDNA analysis was performed using the Illumina platform, NextSeq 2000 System was used with the flow cell P3 (200 cycles ($2\times$ 100 bp) and output 240 Gb).

The HER2 p.G776delinsVC mutation was present with an allele frequency (AF) of 6.05 shortly after the start of T-DXd at the beginning of May 2021 and fell below the assay-validated lower limit of detection of 0.15% AF about four weeks later in June 2021, thus indicating a lack of tumor DNA shedding at this time. From June 2021 to February, HER2 p.G776delinsVC ctDNA levels increased steadily while under T-DXd, reaching a maximum of AF 6.15% in early February, which correlated to clinical progress. In six of the twelve samples, the potential MET resistance alteration MET p.H1112Y was detectable, emerging about two months into T-DXd therapy (Figure 2D). The allele frequency ranged from 0.04 to 0.24% and, thus, was close to the validated LOD of 0.15% of the assay.

Except for the MET p.H1112Y, no further potential resistance mechanism was detected in any of the plasma samples. In addition to ctDNA, carcino-embryonic antigen (CEA) protein levels were determined in blood from the start of chemo-immuno combination therapy in March 2020 throughout three lines of therapy until March 2022. With the start of T-DXd, data were available for both CEA and ctDNA, and the two parameters displayed a high degree of concordance, especially during the first phase after the onset of T-DXd from May through June 2021 (Figure 2A,B).

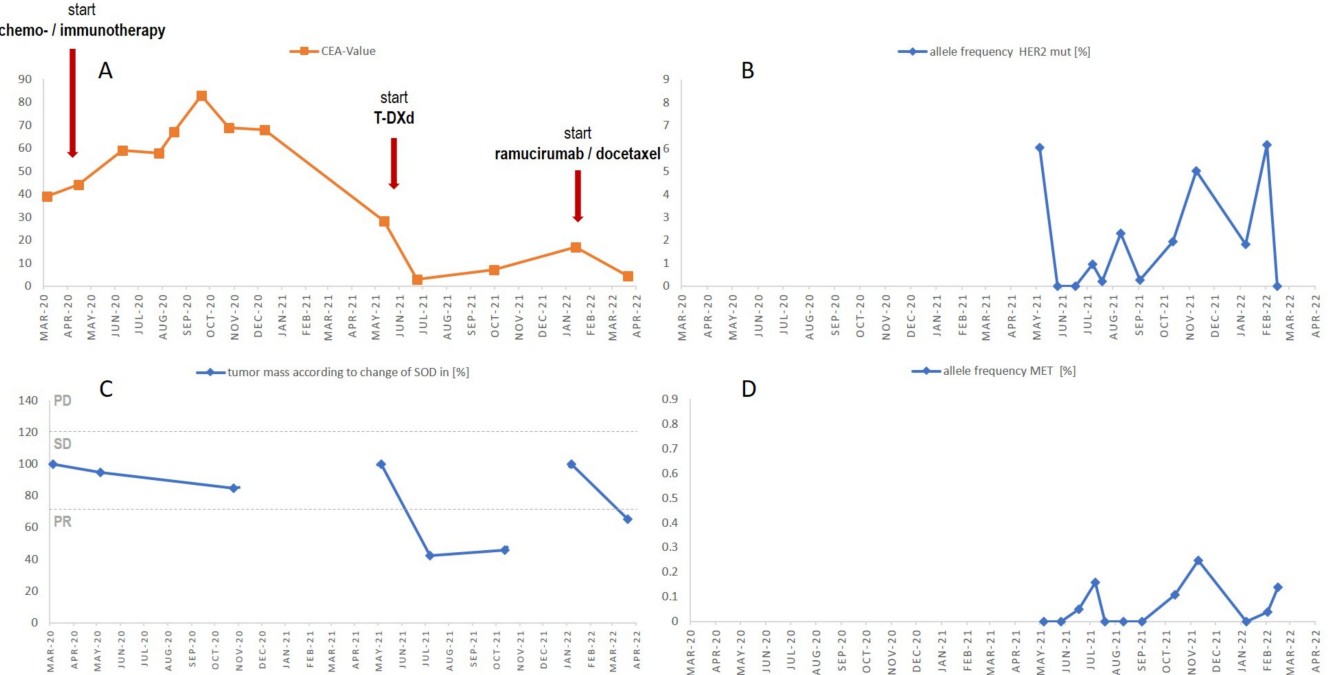

**Figure 2.** CEA levels and change in SOD according to RECIST during 1st-, 2nd- and 3rd-line therapy. In addition, allele frequencies of HER2 and MET mutations are shown during treatment with T-DXd. (**A**) CEA values, (**B**) HER2 p.G776delinsVC allele frequency, (**C**) RECIST 1.1 criteria, change of sum of diameters as % of tumor mass. PD progressive disease, SD stable disease and PR partial response. (**D**) MET p.H1112Y allele frequency. Red arrows indicate change in therapy.

### 3. Discussion

We describe the therapeutic course of a 61-year-old, female HER2 mutation-positive NSCLC patient who responded profoundly but transiently to the antibody–drug conjugate T-DXd administered after relapse to first-line pembrolizumab/pemetrexed/carboplatin. Liquid-biopsy-derived ctDNA levels were used to track the driver HER2 exon 20 insertion and to detect potential resistance alterations. At the time of T-DXd initiation, HER2 p.G776delinsVC was present at relatively high allelic frequency 6.05%, but this decreased after one month of therapy. Over the following six months, the proportion of mutated alleles gradually increased again, and finally reached levels similar to those pre-T-DXd. These kinetics corresponded exactly to the clinical course of disease, however not all liquid biopsy assessments proved successful, for instance the sample taken in September 2021 did not show HER2-mutated ctDNA. Upon reevaluation, this sample showed decreased quality parameter which may be partly responsible for the low allele frequency, however preanalytical issues cannot be ruled out.

The HER2 p.G776delinsVC is a relatively rare mutation in lung cancer but has been previously identified as an oncogenic driver [4]. Most members of the receptor tyrosine kinase ErbB family are activated upon ligand binding. In the case of HER2, in contrast, homo- and heterodimerization with other members of ErbB is the main path to induce downstream signaling, providing a clinical rationale for the use of an HER2-targeted antibody [14]. T-DXd is an antibody–drug conjugate directed against HER2 and is currently being investigated for the treatment of solid tumors in several trials. However, in the unconjugated form, trastuzumab in combination with chemotherapy has yielded disappointing results in NSCLC, reflected by a lack of response independent of the HER2 status [15–17]. During the preparation of this manuscript, the phase II DESTINY-Lung01 trial was published, which enrolled 91 NSCLC patients with HER2 mutations who had been refractory to standard therapy. This showed an objective response rate of 55% for T-DXd and a median duration of response of 9.3 months. Third- or higher-degree drug-related adverse events occurred in 46% of patients, especially neutropenia (19%), anemia (10%) and nausea (9%) [4]. However,

to the best of our knowledge, no data have yet been published on the monitoring of ctDNA during treatment with T-DXd.

Although CEA is not an established biomarker in NSCLC, some reports have supported its role as a sensitive and specific indicator of objective response to chemotherapy and as a predictive marker of outcome during EGFR-TKI therapy [18,19]. CEA levels were monitored throughout the different lines of therapy and their course was concordant with the pattern of ctDNA plasma levels (Figure 2A,B).

Since a limited number of patients have been treated with T-DXd for metastatic NSCLC as of today, little is known about the nature and frequency of potential resistance mutations. Based on experience with trastuzumab maytansine, the first in class agent in breast cancer, a heterogeneous spectrum of on-target and off-target resistance can be expected, including loss of HER2 expression, a kinase switch to HER3 or EGFR, dysregulation of PI3K signaling, or even dysfunctional intracellular trafficking and metabolism [20]. Only a few of the above alterations can be reliably detected by a ctDNA-based hybrid capture NGS. We did detect an MET p.H1112Y point mutation emerging after about two months of T-DXd therapy. The low molecular allele frequencies (0.004–0.25%) make it rather unlikely that this specific MET alteration is the sole contributor to tumor relapse. On the other hand, p.H1112Y is located within the kinase domain of MET and has been defined as activating mutation. In addition, MET alterations have been described as a paradigm mechanism of resistance to targeted therapies; therefore, despite the low AF, we cannot rule out the possibility of a clinically relevant mechanism of resistance and a potential therapeutic target [21–23].

## 4. Conclusions

In summary, we demonstrated that ctDNA monitoring during T-DXd therapy for metastatic NSCLC is feasible and has a good association with tumor response. Disease monitoring of solid tumors is primarily performed by CT imaging and is therefore labor- and cost-intensive. Patients treated with targeted therapy develop characteristic resistance mechanisms on the molecular level. These can be complex and may show subclonal diversity and can be restricted to certain parts of the tumor/metastases. Liquid biopsy testing has the advantage of detecting resistance alterations at the molecular level, ideally before progress is detectable on the imaging level. Although liquid biopsy testing by NGS is currently not reimbursed in Germany, it might represent a future perspective in monitoring disease control of solid tumors. It is important to understand the underlying resistance alterations in detail to adjust next-line targeted therapy. Resistance alterations are heterogeneous and include on-target point mutations or gene amplifications and a wide spectrum of off-target alterations, including translocations, gene amplifications and others [22,24].

**Author Contributions:** Conceptualization, M.F., W.M.B.; methodology, M.F., E.W., S.S. (Stefanie Schmidt)., M.T.; software, M.F., E.W., S.S. (Stefanie Schatz).; validation, M.F., E.W., S.S. (Stefanie Schmidt)., S.S. (Stefanie Schatz)., M.T.; formal analysis, M.F., S.S. (Stefanie Schmidt)., S.S. (Stefanie Schatz).; investigation, all authors; resources, J.H.F., M.G., W.M.B.; writing—original draft preparation, M.F., S.S. (Stefanie Schatz)., W.M.B.; writing—review and editing, all authors.; visualization, M.F., S.S. (Stefanie Schatz)., M.G., W.M.B.; supervision, M.T., J.H.F.; project administration, W.M.B.; funding acquisition, W.M.B. All authors have read and agreed to the published version of the manuscript.

**Funding:** This work was funded by an unrestricted grant to WMB from the "Förderverein des Tumorzentrums Erlangen", FAU Erlangen and the W. Lutz Stiftung, Nuremberg, Germany.

**Institutional Review Board Statement:** The study was conducted in accordance with the Declaration of Helsinki, and approved by the Ethics Committee of the Friedrich-Alexander-University Erlangen-Nuremberg, Erlangen, Germany (protocol number 294 20B from 18 January 2021).

**Informed Consent Statement:** Informed consent was obtained from the patient involved in this off-label use of T-DXd. In addition, written informed consent has been obtained from the patient to publish this paper.

**Data Availability Statement:** The data presented in this study are available on request from the corresponding author.

**Acknowledgments:** The authors thank Ute Herbst and Ramona Paulus for documenting the blood samples obtained.

**Conflicts of Interest:** Wolfgang M. Brueckl has received honoraria for consulting from AstraZeneca, BMS, Boehringer Ingelheim, Celgene, Chugai, Lilly, MSD, Pfizer, Roche Pharmaceuticals and Takeda. Joachim H. Ficker has received honoraria for consulting and/or lectures from AstraZeneca, Bayer, Boehringer Ingelheim, Chugai, GSK, MSD, Novartis, Pfizer, Roche, and Sanofi-Aventis. Markus Tiemann has received honoraria for consulting and/or lectures from Astra Zeneca, Boehringer Ingelheim, BMS, MSD, Novartis, Lilly Oncology, Roche, Takeda. Markus Falk has received honoraria for consulting and/or lectures from Astra Zeneca, Boehringer Ingelheim, Roche, Novartis. All other authors declare no conflict of interest.

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
