# Peer review of "Response of an HER2-Mutated NSCLC Patient to Trastuzumab Deruxtecan and Monitoring of Plasma ctDNA Levels by Liquid Biopsy"

_curroncol, doi:10.3390/curroncol30020130_

Round 1

Reviewer 1 Report (Previous Reviewer 1)

The authors have adequately addressed my comments.

This manuscript is a resubmission of an earlier submission. The following is a list of the peer review reports and author responses from that submission.

Round 1

Reviewer 1 Report

Falk et al report a case of HER2 mutated NSCLC treated with trastuzumab deruxtecan. This is a case which reflects changing practice, with novel HER2 targeted therapy and use of ctDNA dynamics as a marker of response. However, trastuzumab deruxtecan is now FDA approved and ctDNA as a biomarker of response has been demonstrated in many clinical trial cohorts previously – which severely limits the novelty of this case report.

Specific comments

  1. FDA approval for trastuzumab deruxtecan

Given the recent approval of T-DXd, the manuscript needs to be updated to reflect this.

  1. HER2 alterations

The heterogeneity of HER2 alterations needs to be outlined in the introduction in much greater detail. This includes HER2 exon 20 insertion mutations, other HER2 mutations in the kinase domain or other exons in HER2, HER2 amplification and HER2 overexpression on IHC.

  1. NGS assay

More detail is needed on the tissue and plasma NGS assays, such as the panel used, breadth and depth of sequencing etc. Full results of NGS should be provided – especially plasma ctDNA (were other alterations apart from HER2 and MET detected?)

  1. Treatment

Rationale for immunotherapy with pembrolizumab despite PD-L1 TPS 0% should be provided. Method of access to T-DXd should be provided – i.e. on trial, off-label use, compassionate access or other. Toxicities to T-DXd should be provided.

  1. Written informed patient consent

Details of patient consent should be provided.

  1. Resistance mechanism

More detail is needed on the potential MET resistance alteration. Which samples was the MET mutation detected? Was it detected as baseline? Did allele frequency increase over time? This should be included on Figure 2

  1. Figure 2

Y-axis needs labelling. Dotted lines require labelling to indicate meaning. Radiologic findings should also be represented in this figure – such as tumour response/sum of target lesions.

  1. HER2 ctDNA dynamics

Figure 2 suggests fluctuating dynamics, with very low levels in Sep 2021 for example. This warrants discussion

  1. Radiologic response

Response to therapy per CT scans should be determined formally with RECIST v1.1.

Reviewer 2 Report

The report describes a case of a 64-year-old female patient with metastatic HER2 mutated lung adenocarcinoma in whom remission of disease was observed during seven months of second-line treatment with T-Dxd. Additionally, the authors describe the allelic frequency pattern of the HER2 mutation detected by ctDNA analysis and the CEA levels during T-Dxd treatment.

The subject of the efficacy of HER2 targeted treatment and its response monitoring during course of treatment with the use of ctDNA and CEA is relevant and in line with the aims and scope of the journal. However, I have major concerns and believe the manuscript not to be suitable for publication in its current form:

1.      The key messages/gaps in knowledge addressed in the introduction and discussion do not emerge clearly in the current form of the manuscript. More emphasis should be placed on the additional value of the described findings and how these findings should be placed in the current context of NSCLC treatment. Are these findings relevant for treatment decisions in daily clinical practice and why? Which advantages/disadvantages of treatment and treatment monitoring can be expected in the future when applying T-Dxd treatment and ctDNA for response monitoring of T-Dxd treatment?

2.      The manuscript is difficult to read. Readability could be improved by restructuring the introduction and discussion, and by preventing repetition of the provided information on the current literature (e.g. on the efficacy of HER2 treatment options) and word use (e.g. to date (sentence 31 and sentence 32)).

3.      The description of the case (part two of the manuscript) is missing relevant information on the treatment period with T-Dxd and should be more elaborated (e.g. side-effects experienced by the patient, clinical benefit, etc.).

4.      The same radiological response patterns are described with different terms, which makes the visualization of the response pattern more difficult for the reader (e.g. stable remission (sentence 74) and partial remission (sentence 78)). I therefore suggest radiological response patterns to be described by using current RECIST terms (complete remission, partial response, stable disease, and progressive disease).

5.      The figures should be accompanied by figure captions. Furthermore, Figure 2 could be improved by adding the size of the target lesions to visualize the tumor size dynamics during course of treatment.

6.      It is not clear how the MET alteration allele frequency range of 0.04-0.24% does not significantly exceed the validated LOD of 0.15% (sentence 96-97). Since trastuzumab upregulates MET expression, which, in turn, contributes to trastuzumab resistance by cross talk activation of HER3 in breast cancer patients, the finding of the MET alteration should be discussed as a potential mechanism of treatment resistance in the discussion section (important for future interpretation of ctDNA analysis during during T-Dxd treatment).
